# The Cycle to Respectful Care: A Qualitative Approach to the Creation of an Actionable Framework to Address Maternal Outcome Disparities

**DOI:** 10.3390/ijerph18094933

**Published:** 2021-05-06

**Authors:** Carmen L. Green, Susan L. Perez, Ashlee Walker, Tracey Estriplet, S. Michelle Ogunwole, Tamika C. Auguste, Joia A. Crear-Perry

**Affiliations:** 1National Birth Equity Collaborative, New Orleans, LA 20026, USA; awalker@birthequity.org (A.W.); testriplet@birthequity.org (T.E.); 2Department of Social and Behavioral Sciences, University of California, San Francisco, CA 94143, USA; 3Department of Public Health, California State University, Sacramento, CA 95819, USA; 4School of Public Health and Tropical Medicine, Tulane University, New Orleans, LA 70118, USA; 5Department of Medicine, Johns Hopkins University School of Medicine, Baltimore, MD 21205, USA; sogunwo1@jhmi.edu; 6MedStar Washington Hospital Center, Council on Patient Safety in Women’s Health Care, ACOG, Washington, DC 20010, USA; Tamika.C.Auguste@medstar.net

**Keywords:** birth equity, framework, maternal health, maternal morbidity, racial equity, respectful care

## Abstract

Despite persistent disparities in maternity care outcomes, there are limited resources to guide clinical practice and clinician behavior to dismantle biased practices and beliefs, structural and institutional racism, and the policies that perpetuate racism. Focus groups and interviews were held in communities in the United States identified as having higher density of Black births. Focus group and interview themes and codes illuminated Black birthing individual’s experience with labor and delivery in the hospital setting. Using an iterative process to refine and incorporate qualitative themes, we created a framework in close collaboration with birth equity stakeholders. This is an actionable, cyclical framework for training on anti-racist maternity care. The Cycle to Respectful Care acknowledges the development and perpetuation of biased healthcare delivery, while providing a solution for dismantling healthcare providers’ socialization that results in biased and discriminatory care. The Cycle to Respectful Care is an actionable tool to liberate patients, by way of their healthcare providers, from biased practices and beliefs, structural and institutional racism, and the policies that perpetuate racism.

## 1. Introduction

Black birthing people and babies are consistently the most impacted by adverse health outcomes in the United States, and growing literature suggests that experiences of racism and disrespect during healthcare encounters impacts health [1]. Decades of medical and public health research have failed to explain or reduce race-associated differences in maternal outcomes—such as mortality, morbidity, and patient experiences—in the United States. Women of color (i.e., Black, Latina, and Asian women) are more likely to experience a comorbid illness [2] and maternal death [3] and report being unfairly treated within healthcare settings based on their race or ethnicity [4,5]. In addition to increasing trends in maternal mortality overall, perhaps more distressing is the persistent, large, and increasing mortality gaps between Non-Hispanic (NH) Black and all other birthing persons in the United States [3,6]. Healthcare provider factors- delayed response to clinical warning signs, followed by ineffective care- were the most common type of contributor to maternal deaths [5]. An actionable framework that centers Black birthing people, those most impacted by disrespectful treatment and care, is needed to resolve persistent maternal outcome disparities.

## 2. Shifts in Policy and Practice

In wealthy countries like the United States, there is a grassroot and political call to action for a radical shift in practice to reduce inequities in birth outcomes using respectful maternity care as a model for change [7]. The concept of Respectful Care, “care provided to all women in a manner that maintains their dignity, privacy and confidentiality, ensures freedom from harm and mistreatment, and enables informed choice and continuous support during labor and childbirth” [8] is globally accepted. The World Health Organization has called for further research on defining and measuring disrespect in public and private facilities, [5] yet there is not consensus of the ways to improve respectful care.

## 3. Taking Action to Address Maternal Outcome Disparities

This study was developed in response to the growing demand for frameworks to achieve birth equity and promote respectful maternity care in clinical settings that is informed by patients. Current data relies on system needs and does not adequately address the needs of birthing people. In order to address the needs of birthing people, it is necessary to illuminate birthing experiences to inform the ways in which systems might be improved to address disparities in maternal outcomes. The purpose of this study is to create an actionable framework for the practice of respectful maternity care based on the experiences of Black birthing people. This was achieved by eliciting feedback from Black birthing individuals across the United States and incorporating the findings to inform a framework to achieve respectful care. To achieve birth equity and a standard of respectful care, it is time to challenge the frameworks used and the values upheld to determine data collection and quality improvement activities. These measures currently detect symptoms of dysfunction, not the cause [9]. Focusing on easily measurable markers of quality diverts resources from harder-to-measure aspects of care, resulting in unchanged or even worse quality overall [9]. Recent studies support efforts to quantify and codify respectful care as well as address the need for radical change in medical practice [10].

## 4. Methods

In order to ensure the research process centers the experiences of Black birthing people, the research team identified three frameworks to guide the approach for participant recruitment, facilitation, development of the facilitator guide, analysis of the findings, and framework development. Cultural humility [11], Reproductive Justice [12], and Research Justice [13] are the guiding research frameworks of this study. Cultural Humility incorporates a lifelong commitment to self-evaluation and self-critique to address power imbalances in the patient-physician interactions in order to develop a mutually beneficial and non-hierarchical clinical and advocacy partnership with communities on behalf of the individuals and populations [11]. Reproductive Justice is the human right to maintain personal bodily autonomy in an individuals’ decision to have or not have children and to parent children in a safe and sustainable community [12]. Research Justice centers community voices and leadership to be active participants in the process for change and policy reform at local, regional, national, and global levels in order to facilitate lasting social change [13]. Together these frameworks: (a) address the hierarchical structure of medicine, that has historically set aside the needs of the patient, in order to center the needs and experiences of the patient; (b) center the experiences of Black birthing people and the communities they live in; and (c) promote actionable solutions.

## 5. Data Collection

This study was approved by the Institute of Women and Ethnic Studies (IWES) IRB. 

Development of the facilitator guide was informed by an extensive review of literature; feedback from Birthing Justice and Birth Equity activists and researchers, health services researchers, the Institute for Women and Ethnic Studies, and health care providers (e.g., midwives, OB/Gyn physicians, doulas, lactation consultants, nurses, etc.); and Dr. Karen Scott’s Sacred Birth research [14]. All focus groups were co-facilitated by leaders of CBOs serving local communities of Black families.

All participants were given the option of a one-on-one interview to ensure their comfort in sharing their birthing experiences Before the focus group or interview, participants completed a questionnaire that included demographic information, information about utilization of healthcare services, and access to healthcare services. Guiding questions were developed based on a review of the literature and the expertise of study investigators from various disciplines (birth equity research, reproductive justice, health services research, OB/Gyns, and sociology) and tested in a pilot focus group. Modifications were made in the interview guide to decrease the number of questions.

To illuminate the birthing experiences of Black birthing individuals in hospital labor and delivery units, we conducted qualitative focus group discussions and an interview to identify the components of respectful maternity care. All focus groups were moderated and led by two leaders of community-based organizations serving Black birthing communities trained by the research team on facilitation of focus groups and the study research methods. A member of the research team (CLG) was available to the moderators for support and co-facilitation during the focus groups. Focus groups and interviews lasted 2-h.

Childcare was provided and participants were compensated $150 for their time and other study related expenses (e.g., transportation, childcare) related to participation.

## 6. Data Analysis

Digital recordings were transcribed verbatim and reviewed for accuracy in transcription. A team of five (5) scholars with backgrounds and expertise in maternal mental health, Black birthing justice, health services research, and birth equity independently reviewed the transcripts. The team identified overarching themes that were then further refined using codes. Codes were applied to each of the transcripts in a line-by-line coding process. Coders paid specific attention to the role of racism and discrimination in the treatment and care of participants. Themes and codes were developed and identified in relationship to participants’ experiences as Black birthing people. The expansiveness, range of ethnic and cultural groups, of the African diaspora has influenced family lineage in many different countries, and researchers did not want to discount the experience of any birthing person who may inform this inquiry on respectful maternity care.

Data were independently reviewed by three coders. Coders discussed the themes and codes and came to a consensus on recurrent themes, conceptual descriptions, and illustrative examples of respectful care and compiled a codebook for assessing the remaining transcripts. To be included as a salient aspect of respectful care, a theme has to appear in two or more focus groups. Community based organization (CBO) leaders and partners that provide services for black birthing individuals and communities reviewed the preliminary codes and the remaining transcripts were then reviewed.

CBO leaders collaborated with the research team for the initial identification of themes and codes and remained engaged during data analysis. CBO leaders were asked to critically evaluate the codes and draft models in our community validation process. As themes and codes were refined, the research team checked-in with the CBO leaders to ensure the themes and codes were representative of the experiences of their communities and that key themes and concepts were not overlooked. After completion of coding of all the transcripts, the CBOs reviewed the final codes and unique definitions for each of the codes. Five of six CBO leaders refined themes based on the codes and their experiences, refined our definitions, and suggested de novo codes.

## 7. Creating the Framework

Once the themes and codes were established, CBO leaders were asked to provide feedback by walking through several exercises with the research team. The qualitative findings that were identified by the research team, CBO leaders, and stakeholders as critical to respectful care that were then examined in the context of existing frameworks in public health and health psychology. In the style of a listening session, CBO leaders were asked to apply the codes and themes to existing frameworks/models to contextualize the relationships of the codes and themes into the components of a framework to promote respectful maternity care.

## 8. Participant Recruitment

Data were collected in select communities of the United States identified as having higher density of Black births—Atlanta, GA; Baltimore, MD; Chicago, IL; Dallas, TX; Houston, TX; and Tulsa, OK during the Spring and Summer of 2019. Participants were recruited by community-based organizations located in these cities that provide resources and support for Black Birthing individuals in these communities.

The researchers and CBO partners recruited Black birthing people, over 18 years old, who had a birthing experience in a local hospital facility within the last two years. Blackness and identifying as Black includes a wide range of ethnic diversity, including African immigrants, Indigenous groups, Afro-Latinx, mixed race, etc. It was important that participants self-identified as Black and Blackness is a personal identity. In the participant demographic questionnaire, they were all asked to identify their race and gender identity. As all participants identified as Black and as women, they will be referred to as Black mothers for the duration of the report.

## 9. Framework Development

The themes and codes were applied to develop a framework for achieving respectful maternity care with the feedback and input from stakeholders (e.g., clinicians, nurses, birth workers, professional organizations, and health service researchers) and leadership from community-based organizations. After the identification of codes and themes, the research team and CBO leaders underwent an iterative process to develop a framework that were actionable steps for providers to practice respectful maternity care that included the themes from the focus groups. See Figure 1: Methods Process for an overview of this methodological approach.

## 10. Results

Seven focus groups with 3–11 participants per focus group and one (1) one-on-one interview (*N* = 50) were held with Black mothers. Participants represented numerous communities across the United States and a range of socioeconomic backgrounds. See Table 1: Participant Demographics and Characteristics.

The research team explored and presented numerous existing models and frameworks including the Bronfenbrenner Ecological Model [15], the Feminist Ecological Model [16], and Maslow’s Hierarchy of Needs [17] to stakeholders and CBO leaders in order to provide context for how the themes and codes might fit into existing frameworks and models.

CBO leaders unanimously rejected the idea of a hierarchical framework to achieve respectful care, noting that the absence of respect does not prevent a birth from happening, such that there is no one value or event that leads to a respectful birth experience. CBO leaders identified a need for a model of respectful care that was cyclical and relational, rather than hierarchical. The CBO leaders asserted that a linear process is irrelevant in communicating how providers achieve respectful care and that respectful care is dependent on antiracism and eliminating individual racist beliefs. They unanimously accepted the depiction of the cyclical model that promotes ongoing growth toward respectful care practices by confronting racism.

A CBO leader introduced the research team to Bobby Harro’s Cycles of Liberation [18] and Socialization [19]. Examining all the existing frameworks and models, all research partners arrived at a consensus that a cyclical model best illustrates the continuous and ongoing work of achieving respectful care. The Cycle of Liberation was adapted to include focus group themes to arrive at the Cycle to Respectful Care.

Themes and codes identified from the focus groups were incorporated into the Cycle to Liberation. (See Figure 2: Cycle to. Respectful Care) To fit the Cycle to Liberation with the focus groups and interview findings, the research team used an iterative process engaging CBO leadership, Birthing Justice and Birth Equity activists and researchers, health services researchers, birth equity researchers, leaders at professional organizations, and health care providers (e.g., midwives, OB/Gyn physicians, doulas, lactation consultants, nurses, etc.) to integrate the themes and codes into, what is now, the Cycle to Respectful Care. In addition to the feedback from key stakeholders, the research team identified existing quality improvement initiatives and models of care to incorporate into the framework to provide actionable solutions. The iterative process led to the Cycle to Respectful Care.

## 11. The Core of the Cycle to Respectful Care

The core connects to each section of the cycle. Therefore, as a person progresses around the cycle the core values remain. Some of the values are present when a person first enters the cycle, those values are then fostered, expanded upon, and matured as they proceed through the various phases. The core values of the Cycle of Respectful Care include valuing patients’ Black Intersectionality [1,5], Birth Equity, Reproductive Justice [12,20], the Professional Pledge/Oath [5] healthcare professionals commit to at the start of their career, Holistic Maternity Care, Humanity [21,22,23], and Love of self and others. (See Table 2: Definitions and Descriptions of the Core.) Values are strengthened through each phase, as they exist and operate on both the individual and collective levels. Harm can come from anywhere, but for the purpose of illustrating the ways in which the cycle could be put into practice Table 3: Examples of Individuals Moving through the Cycle illustrates the process by which a physician, nurse, and patient might move through the cycle. See Table 4: Moving through the Cycle to Respectful Care for the rationale and evidence for each phase.

## 12. Discussion

Historically, research in underserved communities has not included community partners, instead it has made these communities serve as laboratories, and the community members as experimental subjects [37]. This study engaged birthing communities, stakeholders, and organizations, which allowed for unique insight about specific needs and health impacts observed by Black mothers. Participants’ social and cultural experiences informed our study methods, research questions, research findings, and the iterations in the development of the Cycle for Respectful Care. Active and informed participation of birthing people in all aspects of the design and implementation of institutional change was critical for constructive accountability [1,38].

Qualitative findings illuminated the hospital birthing experiences of Black birthing people to inform an actionable framework for healthcare providers and birth workers to address persistent and widening disparities in maternal health outcomes. Engagement with stakeholders at all steps of the research process ensured fidelity to centering the experiences of Black birthing people. An iterative process to identifying a framework that appropriately contextualizes the themes and codes of the qualitative findings resulted in the Cycle to Respectful Care—an actionable framework to address biased and disrespectful care. A high level of trust was established and investment in the process was mutual.

Accordingly, the Cycle to Respectful Care framework combines theory, analysis, and experiences of Black mothers across the United States It describes a recurrent process that is reminiscent of successful social change efforts, which led to some extent of liberation from the negative impact of oppressive systems for everyone involved. The Cycle to Respectful Care supports intentional patient engagement, promotes advocacy, and is an actionable tool to address birth equity in maternity care. The Cycle to Respectful Care framework was created for health systems and providers as an actionable guide toward respectful care for Black mothers, and eventually all birthing people.

The Cycle to Respectful Care is a tool for health care providers to understand how their Black patients experience disrespect, and how patients could be liberated from biased practices and beliefs, structural and institutional racism, and the policies that perpetuate racism. The primary provider audience for this framework is physicians, nurse midwives, and nurses because participants’ hospital and clinic experiences are highly influenced by these individuals. Any care provider can operate in a disrespectful practice that may or may not be rooted in racism. Though this framework is written for the direct use of medical professionals, it can be used by any health care professional who serves Black birthing people in their reproductive life course.

The Cycle to Respectful Care flows through the levels of racism [39], internalized racism, personally mediated racism, and institutionalized racism [40]. Harro [18] claims that a person can start anywhere on the Cycle of Liberation, but that it is usually inevitable that intrapersonal, interpersonal, and systemic change will transpire while using this framework, which causes one to “wake up”. After entering, one may repeat the process many times as there is no end to pursuing equity.

The Cycle to Respectful Care theoretical framework is based on the birth experiences of Black mothers in order to inform the ways in which to achieve respectful care. This framework complements existing provider educational tools, promotes anti-racist and birth equity practices, and bridges community assets to hospital care by centering the cultural, biopsychosocial, and holistic needs of Black mothers in order to reduce disparities in clinical and patient reported experience measure outcomes for all birthing people.

Limitations of this study were that one of the six CBO leaders was not able to participate in the development of the framework and validation of the qualitative analysis. Other limitations were that the communities selected for recruitment were based on a convenience sample of the areas where the research team had established relationships. This resulted in a lack of geographic spread that excluded middle America and rural communities.

As hospitals and health systems look for solutions to promoting equity and resolving disparities, this framework is a guide for hospital staff and administrators who seek to achieve equity in their practice. The development of the Cycle to Respectful Care framework will ensure that hospital staff have tools to introduce and continuously work toward respectful care practices. As tools to address the patient led development and validation of measures for respect and discrimination in maternity care progress, a clinical standard for community engages quality improvement will emerge. From the perspective of professional responsibility and anti-racism, it should be expected that healthcare providers prioritize frameworks created for, by and with Black birthing people.

## 13. Conclusions

The Cycle to Respectful Care theoretical framework underscores the process to achieving respectful care for Black mothers and all birthing people. This framework is the inception of Respectful Maternity Care provider education tools, community awareness messaging, and a patient reported experience measure. The framework is expected to have applications in healthcare quality improvement initiatives for improving experiences and outcomes for birthing people (Appendix A see PREM FACILITATOR GUIDE).

## Figures and Tables

**Figure 1 ijerph-18-04933-f001:**
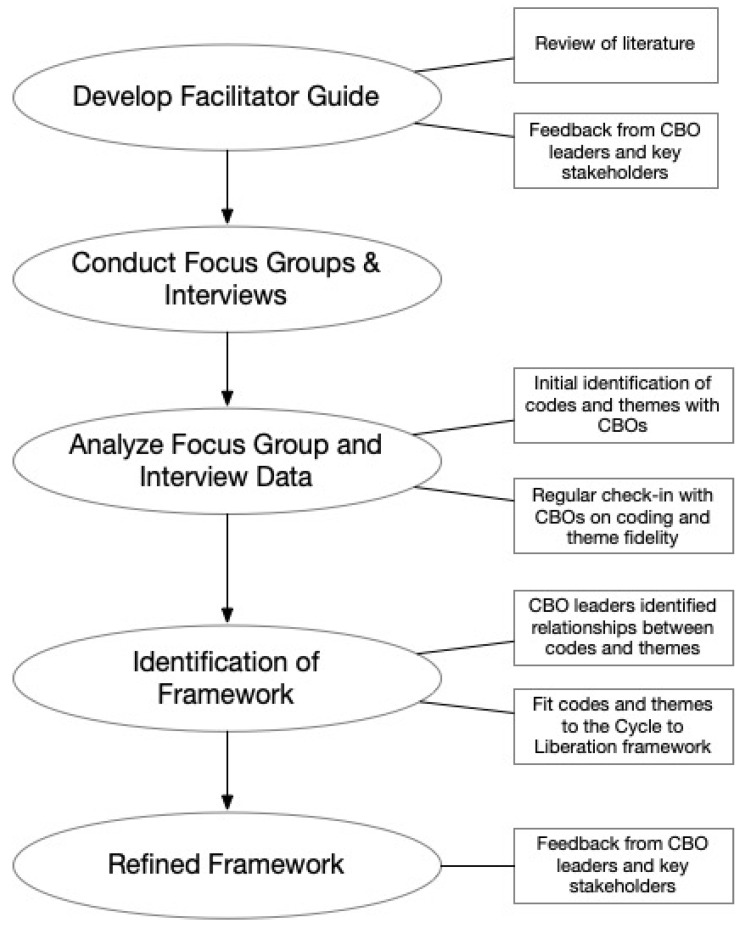
Methods Process. CBO: Community based organization.

**Figure 2 ijerph-18-04933-f002:**
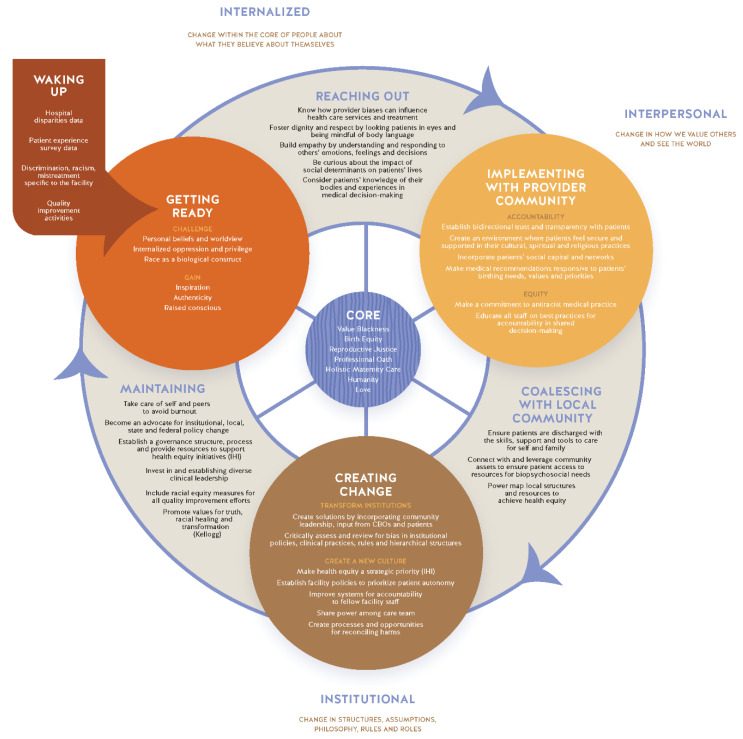
Cycle to Respectful Care.

**Table 1 ijerph-18-04933-t001:** Participant Demographics and Characteristics.

Variable	*n* (%)
Location	
Atlanta	11 (22%)
Baltimore	7 (14%)
Chicago	9 (18%)
Dallas	6 (12%)
Houston	9 (18%)
Tulsa	8 (16%)
Insurance	
Public	25 (50%)
Private	16 (32%)
Both (Public and Private)	7 (14%)
Decline to state/Unsure	2 (4%)
Annual Household Income	
<$25,000	16 (32%)
$25,000–$49,999	18 (36%)
$50,000–$74,999	7 (14%)
$75,000–$99,999	4 (8%)
$100,000+	5 (10%)
Education	
Some High School	3 (6%)
High School Graduate/GED	6 (12%)
Trade School	1 (2%)
Some College	17 (34%)
Bachelor’s Degree or Higher	23 (46%)

**Table 2 ijerph-18-04933-t002:** Definitions and Descriptions of the Core.

Core Value	Description
Black Intersectionality *	Valuing the Black experience rather than the physical dark skin. Maternal experiences from Black identifying mothers are rich with data about bias and racism negatively affecting their births. It is imperative that making strides in quality improvement efforts to value the culture and experiences of being Black. Any quality improvement in maternal experiences are hypothesized to impact Black women most directly; therefore, any solutions developed must explicitly center Black women [1].
Birth Equity *	The assurance of the conditions of optimal births and wellbeing for all people with a willingness of systems to address racial and social inequities in a sustained effort.
Reproductive Justice	Capacious envisioning of reproductive possibilities that requires the use of intersectionality, the perspective that allows us to comprehend how race, class, ethnicity, and sexuality together construct gendered implications of motherhood and citizenship, sex and reproduction [12,20].
Professional Pledge/Oath	The commitment and promise of each profession. This is included in the core to remind hospital staff of the reasons why they practice and the foundational values of their profession. This could include the Hippocratic Oath, Imhotep Oath, Nightingale Pledge, etc. [5].
Holistic Maternity Care	Black Mamas Matter Alliance’s (BMMA’s) holistic maternity care concept is anchored in: addressing gaps in care and ensures continuity of care, is confidential, safe and trauma-informed, is culturally informed and includes traditional practices, respects spirituality and spiritual health, and lastly is provided by culturally competent and culturally congruent providers [21,22].
Humanity	Characterized by the United Nations treaty for Human Rights. From the perspective of mothers, being treated with humanity is being seen and regarded equally on the same level as another person you are interacting with, kindness, courtesy and politeness [23].
Love of self and others *	Respectful care is the practice of love. It is developing a sense of self as a care provider so that they can love others who are different than themself.

* Definitions based on the focus group findings.

**Table 3 ijerph-18-04933-t003:** Examples of Individuals Moving through the Cycle.

Cycle Stage	Physician	Nurse	Black Patient
Waking-Up	A physician does not believe themselves to be personally are racist, but the data from their medical director shows disparities in HCAHPS (Hospital Consumer Assessment of Healthcare Providers and Systems) [24] and C-section rates for Black patients.	A nurse views countless news stories on police brutality and the Black Lives Matter Movement, and thus, she begins to reflect upon her role in contributing to racism.	A patient realizes she gets different types of questions from her doctor than the White mothers in her mom group. Questions such as whether she plans to terminate or continue the pregnancy and questions about her relationship with the baby’s father.
Getting Ready	A physician is required to take implicit bias training but is uncertain on how the training will impact their medical practice.	A certified nurse midwife (CNM), continues to raise her consciousness and educate herself on privilege and the construction of racism in the U.S.	A patient begins to educate herself on her birthing options and the hospital policies.
Reaching Out	A physician recognizes the ways in which their biases influence patient care and seeks to identify ways in which their practice might be more holistic by asking patients about their experiences at home, at work, and with family.	A nurse midwife could start with practicing new approaches with her patient interactions, such as looking patients in the eyes when she’s speaking with them, showing patients they have her full attention with the positioning of her body, and making a conscious effort to listen to patients while checking her own biases.	A patient communicates her birthing needs and priorities with her care team.
Implementing with the Provider Community	A physician is, perhaps, now aware of the patient’s support system and considers the patient’s knowledge of their body in medical decision-making.	A nurse midwife becomes her department’s champion for educating the staff on best practices for accountability and decision-making.	A patient asserts her knowledge of her body and experiences to create a birthing plan where she would feel most safe and supported.
Coalescing with the Local Community	A physician ensures patients are discharged with all that they need to care for themselves and their family by connecting with and leveraging community assets.	A nurse midwife leads a power mapping exercise, starting with her network of local CMNs, to identify structure and processes for health equity.	When a patient shares her birth plan, the nurse provides resources to complement the birth plan and to meet the patient’s biopsychosocial needs.
Creating Change	A physician might suggest at quality improvement meetings with all hospital staff to create a system of accountability and leveled hierarchies among all hospital staff.	A nurse midwife builds relationships with local Women, Infants, and Children (WIC) offices and CBOs and creates a transparent process for patients to report harms, mistreatment or complaints.	A patient is educated on ways to report harms and complaints, and they are invited to participate in a department-wide maternal mortality review committee as a patient liaison.
Maintaining	A physician advocates for institutional policy and on-going workshops/medical education to minimize risk of burnout.	A nurse midwife engages with her statewide professional organization to establish policies for investment and promotion of diverse hiring practices.	A patient is introduced to services at health systems and hospitals that have shown a commitment to racial equity, made possible by the strength of community-hospital partnerships

**Table 4 ijerph-18-04933-t004:** Moving through the Cycle to Respectful Care.

Cycle Phase	Definition	Actions
Waking Up	In the Cycle to Respectful Care, this waking up might include a critical incident of racism, discrimination, or mistreatment in the healthcare facility. Providers are made aware of the incident through patient reports, disparities data, or mandates to address disparate outcomes through implicit bias training or a quality improvement initiative. The American College of Obstetrics and Gynecologists (ACOG) ACOG AIM Patient Safety Bundles [25] and quality improvement initiatives are resources for hospitals and health systems that may illuminate racist and discriminatory care in order to prompt this Waking-Up phase.	Due to the urgency of growing maternal inequities in the United States, hospitals, healthcare systems, and policy makers have taken action to mandate implicit bias training thereby initiating the Waking-Up process rather than waiting for maternal care staff to Wake-Up on their own. The Cycle to Respectful Care begins when an individual observes or experiences the world differently than s/he has in the past.
Getting Ready	Getting Ready is the point at which individuals move from exclusively internal work to application in how they interact with and speak to others. This can be achieved through reviewing evidence-based research, attending anti-racist workshops, training on various topics, and building connections with others. This phase can include challenging beliefs in our worldview, medical education, and consciousness raising.	In preparation of the practice of valuing Black mothers more intensely, healthcare providers become conscious and make note of thoughts, language, and actions to see if they are consistent with newly established beliefs or they can be dismantled [18]. This intrapersonal section requires us to develop a repertoire of skills and tools that will serve us throughout the rest of the Cycle. These skills are built with others in Reaching Out.
Reaching Out	Reaching Out describes the ways in which an individual solidifies a new understanding of Respectful Care. Providers, who are educated about their biases, can identify the behaviors they exhibit that influence care and treatment for patients they are biased towards. Communication and information sharing become more important to the provider, to show themselves more trustworthy [5]. Overall, curiosity about the impact of social determinants on patients’ lives can help providers build empathy and understand others. Small shifts in behavior communicate a level of respect to Black mothers, that impacts their experience. This Reaching Out phase moves us from the internal work to the truly interpersonal within the provider community.	In this phase, a person begins to incorporate their new ideals and knowledge into their everyday interactions, observing the response of others in their life to their new perspective. It is imperative to practice the new skills with others, test expressing new views, vocalize uncertainty instead of staying silent, and examine ideals through reflection and introspection, and seek out a greater range of differences than before. This Reaching Out phase provides strategies to practice the ways in which new worldviews will be met by patients.
Implementing with Provider Community	The Provider Community phase of the Cycle to Liberation contains two components: conversing with those who possess similar social identities and those who are different to build coalitions [18]. The interpersonal phase of the respectful care process is marked by a change in the value of others. This phase is characterized by the creation of an ongoing dialogue where views are exchanged, people are listened to and valued, and the process of seeing others’ points of view as making sense and having integrity, even if they are very different from our own, begins. An integral part of this dialogue is exploring differences, clarifying them, erasing assumptions, and replacing them with firsthand contact and good listening. Building a solid provider community for respectful care means that small groups within institutions come to a common understanding of respectful care for Black mothers. These individuals will begin to conduct themselves differently in ways that impact their colleagues.	Addressing the provider community consists of two steps: (1) dialoguing with people who are like us for support and (2) dialoguing with people who are different from us for gaining understanding about oppression. Patients’ culture, religion, fears, and hopes for their birth experience, must be discussed. It is useful to collectively create guidelines on normalizing best practices that are not standardized, like visitation policies or emerging anti-racism tools [26]. The Black mothers in the study suggested that provider accountability is shown either to the health system itself or to the patient. It is socialized in the provider community to build trust with the patient, such that the patient adheres to clinical guidance. It is not socialized in the provider community to be responsive to the knowledge, words, birthing needs, and priorities of the patient as an autonomous decisionmaker in the birthing experience [27]. The Institute for Healthcare Improvement (IHI) Framework [28] and AIM Patient Safety Bundle Reduction of Peripartum Racial/Ethnic Disparities are resources for organizational self-evaluation for accountability and equity of care to guide health systems and hospitals through this phase.
Coalescing with Local Community	The Coalescing phase is where the actions of the organized coalitions and groups begin to disrupt oppressive systems and create change [18]. It is vital during this phase to realize the collective work of the cohesive group is greater than individual actions. Once obstacles have been ameliorated, one can address factors that maintain racial inequities by joining forces with the broader community beyond that of the hospital facility or healthcare system [27].	Working in a true collaborative manner means that providers are culpable in ensuring patients are well when they are outside of the direct care and oversight of the care facility. Providers, who have coalesced with their patients’ communities are able to ensure patients leave with access to resources to meet biopsychosocial needs. This may require large systems to power map structures and processes for health equity in their locations. This phase is intended to disrupt the status quo and for members of coalitions to take a stand with their beliefs. Consistent with existing quality improvement efforts, this phase also aligns with the IHI Framework sections four and five [28], National Quality Forum’s domains of health equity [27] and the ACOG AIM Patient AIM Patient Safety Bundle Reduction of Peripartum Racial/Ethnic Disparities [29]. The actions in this phase of the Cycle lead to collective work to create change in structures, assumptions, philosophy, rules, and roles
Creating Change	The Creating Change phase of the cycle includes redesigning health services to create new culture and norms that reflect the public’s collective identity [30], resulting in new assumptions, new structures, new roles, and new rules consistent with birth equity. Establishing health equity as a strategic priority and challenging structures, greatly enhances efforts to critically transform systems. Another way to create a culture of respectful care is to improve methods of accountability so that providers can have critical conversations amongst themselves, this involves taking leadership risks and becoming a beacon of change. Critical transformation takes place when organizations make conscious collective decisions for all policies for a collaborative structure rather than hierarchical. The new assumptions, rules, roles and structures must be cultivated.	*Creating Change* for respectful care means transforming institutions and creating a new culture [31]. Centering the voices and experiences of the group most impacted by the inequity is of greatest importance. Institutional change also depends on an assessment of institutional policies, clinical practices, and structures undergirding the system. Mentorship from identified birth equity champions within the facility or dismantling structures of power in and around quality improvement projects are legitimate activities to create change [5,32]. Tools for assessments include the Institute for Healthcare Improvement tools, National Quality Forum’s health equity domains [27], AIM tools and the strategy of applying the Brooks Equity Typology assessment [33] described in the R4P framework [34].
Maintaining	Maintaining is a phase where all the previous changes become routines in the life of the person, and that people in this phase of the system support each other, to hold one another accountable for maintenance of the change [18]. Moving towards a respectful and anti-racist practice necessitates that institutions challenge structural racism and other intersecting oppressive systems (e.g., ableism, classism, ethnocentrism, homophobia, sexism, transphobia- and shift power in resources, leadership, and policies) [10]. In order to succeed, change needs to be strengthened, monitored, and integrated into the ritual of daily life. Just like anything new, it needs to be nurtured, learned again “debugged,” and modified as needed. Maintaining includes a paradigm shift in medical training and competencies for respectful and anti-racist care models [10]. There must be an ongoing commitment and investment of resources from leadership and systems to maintain Respectful Maternity Care initiatives.	Providers are under extreme pressure and responsibility. Taking care of themselves and others on the care team helps them to avoid burnout and desensitization from repeated issues. Individual providers can help maintain systems change by advocating for institutional, local, state, and federal policies that impact social determinants [30], structural racism [35,36] and healthcare overall. They may be trained to advocate or exercise their inherent knowledge of the administrative or legislative process. Advocacy within the healthcare facility would be helping establish a governance structure and process for health equity, including hiring a diverse staff. Quality improvement efforts of any kind must address racial equity and bias to impact care. The Institute for Healthcare Improvement Achieving Health Equity: A Guide for Health Care Organizations provides guidance on this process [28].

## Data Availability

The data presented in this study are available on request from the corresponding author. The data are not publicly available due to data that may potentially compromise participant anonymity.

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
