# Peer review of "The Cycle to Respectful Care: A Qualitative Approach to the Creation of an Actionable Framework to Address Maternal Outcome Disparities"

_ijerph, 2021, doi:10.3390/ijerph18094933_

Round 1
Reviewer 1 Report
Reviewer statement:
The Cycle to Respectful Care: An Actionable Framework to Address Maternal Outcome Disparities
Increased mortality and morbidity are still reported in women of color despite of medical and public health research. The authors have conducted a study to understand birthing experiences with labor and delivery in hospital setting, by eliciting feedback from Black birthing individuals in order to inform a framework to achieve respectful care. This knowledge can provide valuable information and guidance for the management of color women throughout their pregnancy and delivery, but also in health care in general, which is relevant in clinical practice. This information is important to health care professionals and could guide further knowledge, provision an care and research on this topic.
Title: The title reflects the topic being investigated.
- The authors could considered adding detail of the study method to the tiltle, e.a based on focus group discussion and interview.
Overall: The paper is well written. The English grammar and style are fine throughout the entire article, minimal revision is required.
Abstract : see overall remarks and remarks throughout the article.
Introduction:
The introduction section is attractive to read form a reader point of view, explaining the reason and purpose for conducting this study. The length of the introduction section is considered excellent.
- In the section: Taking Action to Address Maternal Outcome Disparities, the authors report details of the method used to conduct the study, lines 65-69: “ The purpose of this study is to understand birthing experiences with labor and delivery in the hospital setting by eliciting feedback from Black birthing individuals across the United States in order to inform a framework to achieve respectful care. “ The details: eliciting feedback from Black birthing individuals across the United States should also be mentioned in the method section, please do so.
Methods :
This section is attractive to read form a reader point of view, explaining the methods used to conduct this study. Despite there are some points needing explanation and/or clarification.
- As I was reading the article, I got a bit loss in the different process steps conducted to realize the framework. The authors could help the reader in understanding the different steps in recruitment of information and establishing the framework by using a flow chart, showing the an overview of the process. This helps the reader in understanding the different steps which improves the understanding of the article. This could be provide as a supplemental figure.
- The authors report in line 79-80: “ Qualitative focus group discussion and interviews were conducted to identify the components of respectful maternity care.” The authors should provide more details on this crucial step in the process of framework development. Please do so?
- The recruitment process of Black birthing people is not explained , please do so as this can introduce bias.
- The authors report in line 114-119: “ Development of the facilitator guide was informed by an extensive review of literature; feedback from Birthing Justice and Birth Equity activists
and researchers, health services researchers, the Institute for Women and Ethnic Studies, and health care providers (e.g., midwives, physicians, doulas, lactation consultants, nurses, etc.); and Dr. Karen Scott’s Sacred Birth research [13].” Feedback by gynecologists are not reported, an important medical specialist in the field of birthing. Could the authors explained why this was not done?
- In the section : Analysis and Framework Development in line 128-129 the authors report: “ Cultural humility [14], Reproductive Justice [15], and Research Justice [16] are the guiding research frameworks of this study.” As a reader this information came out of the blue. Why did the authors chose these items as the research frameworks for this study?
- And how does this relate to the facilitator guide?
Results:
This section was good to understand and easy to read. Despite, some minor points need explanation and clarification.
- I suppose that all participants received a one-on-one interview?
- The authors report only 6 location, were this the only 6 location in which women were recruited?
- Why only these locations?
- When looking a t the aim of the study, see line 65-69, “ understanding birthing experiences with labor and delivery in the hospital setting by eliciting feedback from Black birthing individuals” this result is not presented in the result section. Please elucidate on this point.
- 68 across the United States in order to inform a framework to achieve respectful
- 69 care.
- In line 155-159 the authors report: “ The research team then examined the themes of respectful care in the context of existing frameworks in public health and health psychology. We began with the Bronfenbrenner Ecological Model [17], the Feminist Ecological Model [18], and Maslow’s Hierarchy of Needs [19].” From a reader point of view I do not understand why this was done in this face of the study. As reported earlier by the authors a broad group was consulted, please explain this to the reader. Again the use of a flow diagram can be helpful.
- The authors now introduce CBO leaders as important actors in the process. They critical evaluated the codes, drafted models, provided feedback. Furthermore, the rejected the idea of a hierarchical framework and had more impact of the study. The input of the CBO leaders was enormous which for me as a reader raised question on the method use, could the authors please elucidate on the raised conclusion?
- Figure 1 was not available and could not be reviewed.
- The description of the tenants of the cycle to respectful care are far too long. This should be shortened to make it attractive to the reader. Furthermore the use of supplement material should be considered.
Discussion
The discussion section is attractive to read, but much too short.
- In the discussion section the authors should provide an overview of the results of the study conducted, with the perspective of the available literature on this topic. Please do so.
Tables:
No comments
Reviewer 2 Report
OVERVIEW
This manuscript reports of development of a document intended to provide actionable framework to address maternal outcome disparities. Focus group sessions were held with Black mothers who have recently given birth for input, but the data from the focus group are not provided (just demographic characteristics of the participants). How the information from the focus group were used to transform into the final document is not clear either. Terms like “codes”, “themes”, “components”, and “models” are used, but it is not possible for this reader to understand the process. Documents like this are important, but I don’t think publishing this one as an original article is suitable. Perhaps editorial or commentary may be more suitable (?)
SPECIFIC COMMENTS
TITLE
The title needs to be re-written. As is it is misleading as it seems to indicate this framework has already been implemented. It may be better to add “Creation” or “Development of” after the colon.
ABSTRACT
The abstract needs to be re-written. It describes what “The Cycle of Respectful Care” is, but does not describe that this manuscript is about its development.
INTRODUCTION
Lines 66-70. It is stated that “The purpose of this study is to understand birthing experiences with labor and delivery in the hospital setting by eliciting feedback from Black birthing individuals across the United States in order to inform a framework to achieve respectful care.” However, the paper does not report on these experiences, and is focused on describing “The Cycle of Restpectful Care” framework.
METHODS
What the authors mean by “components”, “codes”, “unique definitions”, “themes” etc. and how they differ is not clear.
Lines 87-90. This sentence starting with “Active and informed….” Should be moved to Discussion.
Line 98. Would it be more accurate to use “Subject” rather than “Sample”?
Conflicting information on inclusion criteria. Lines 101-103 says “The researchers considered a wide range of ethnic diversity to be considered Black,….”; however, lines 107-109 says “Blackness is defined in this study as a personal identity, therefore, researchers thought it was most important that participants self-identified as Black.”
Lines 103-108. The sentence starting with “The expansiveness of the African…” needs to be re-worded for clarity. Providing an example may be useful.
Line 114. The facilitator guide is not available for review.
How focus group data were recorded is not mentioned.
RESULTS
Lines 146-149. The content of two consecutive sentences starting with “Seven focus groups…” should be divided into Methods and Results.
Lines 155-158. The two consecutive sentence starting with “Focus group findings…” should be moved to Methods.
Lines 193-197. The paragraph starting with “The Cycle of Respectful Care….” Should be moved to Discussion.
Reviewer 3 Report
The manuscript ijerph-1165420 entitled with "The Cycle to Respectful Care: An Actionable Framework to Address Maternal Outcome Disparities" written by Dr. Carmen L Green has been submitted to IJERPH. Overall, it's an interesting study to me. However, the format of manuscript should be modified to follow the journal's style before it can be accepted for publication.
Round 2
Reviewer 2 Report
Please see the attached file.
